# Various Perspectives on Microbial Lipase Production Using Agri-Food Waste and Renewable Products

Tomasz Szymczak [1,2], Justyna Cybulska [1,*], Marcin Podleśny [2] and Magdalena Frąc [1]

[1] Institute of Agrophysics, Polish Academy of Sciences, Doświadczalna 4, 20-290 Lublin, Poland; Tomasz.Szymczak@grupaazoty.com (T.S.); m.frac@ipan.lublin.pl (M.F.)

[2] Azoty Group Zakłady Azotowe "Puławy" S.A., al. Tysiąclecia Państwa Polskiego 13, 24-110 Puławy, Poland; Marcin.Podlesny@grupaazoty.com

[*] Correspondence: j.cybulska@ipan.lublin.pl

**Abstract:** Lipases are enzymes that catalyze various types of reactions and have versatile applications. Additionally, lipases are the most widely used class of enzymes in biotechnology and organic chemistry. Lipases can be produced by a wide range of organisms including animals, plants and microorganisms. Microbial lipases are more stable, they have substrate specificity and a lower production cost as compared to other sources of these enzymes. Although commercially available lipases are widely used as biocatalysts, there are still many challenges concerning the production of microbial lipases with the use of renewable sources as the main component of microbial growth medium such as straw, bran, oil cakes and industrial effluents. Submerged fermentation (SmF) and solid-state fermentation (SSF) are the two important technologies for the production of lipases by microorganisms. Therefore, this review focuses on microbial lipases, especially their function, specificity, types and technology production, including the use of renewable agro-industrial residues and waste materials.

**Keywords:** microbial lipase; agri-food waste; submerged fermentation; solid-state fermentation

## 1. Structure and Specificity of Lipases

Lipases or triacylglycerol acyl ester hydrolases belong to the serine hydrolase family and they are known as carboxylic acid esterases (EC 3.1.1.3). The first lipase use was investigated by Claude Bernard in 1848. His first major scientific discovery showed that the pancreas produces a substance capable of emulsifying and saponifying fats. This discovery not only explains part of the exocrine function of the pancreas, but also the mechanisms of the digestion and absorption of fats [1]. Lipases can be produced by a wide range of organisms, including animals using organs like the pancreas, liver, stomach and intestinal wall, and also by plants using seeds and vegetative organs, as well as by microorganisms [2]. Microorganisms, such as bacteria [3], fungi [4] and yeast [5], have the ability to produce intracellular, extracellular or membrane-bound lipases. Furthermore, microbial lipases are more stable, they have substrate specificity and a lower production cost as compared to other sources of these enzymes [6–8]. The first microbial lipases were isolated by Eijkmannin [9] from *Bacillus prodigiosus*, *Bacillus pyocyaneus* and *Bacillus fluorescens*, which are currently known as *Serratia marcescens*, *Pseudomonas aeruginosa* and *Pseudomonas fluorescens*, respectively.

Lipases are built on an α/β hydrolase fold, which is composed of a core of predominantly eight parallel β-strands forming a super-helically twisted central β-sheet surrounded by a varying number of α-helices. Moreover, some variations in the α/β fold were found in several lipases and consist of differences in the amount of α-helices or β-sheets and loop lengths [10]. The main catalytic site of lipases is the moiety, which contains the G-X-S-X-G sequence, where G = glycine, S = serine, X = any amino acid. This catalytically active moiety of lipase was found not only in lipases, but also in other hydrolytic enzymes, such

as serine hydrolases [11]. The key structural components of lipase include the lid, binding pocket, oxyanion hole and disulfide bond. The lid is composed of one or more α-helices, which are connected to the main structure of the enzyme by a flexible structure. The lid is a mobile element and it can open the active site in the presence of a biphasic system reaction. When the lid uncovers the active site of the enzyme, it allows for access of the reaction substrate [12]. The binding pocket is the active site of the lipases, it is located on the top of the central β-sheet of the protein structure. The substrate-binding sites of lipases vary in their size, shape, the deepness of the pocket and the physicochemical characteristics of their amino acids. The following three types of binding pocket geometry may be distinguished: hydrophobic crevice-like, funnel-like or tunnel-like. Lipases from *Rhizopus* and *Rhizomucor* have a hydrophobic, crevice-like binding site, which is located near the surface of the protein. Lipases produced by *Burkholderia* spp. have a funnel-like binding site and, for example, lipases obtained from *Candida rugosa* have a tunnel-like binding site [13]. The oxyanion hole influences the catalytic efficiency of the enzyme. This important component has a large influence on hydrolysis because during the reaction, the oxyanion hole stabilizes the charge distribution and reduces the state energy of the tetrahedral intermediate by forming hydrogen bonds. Lipases are also characterized by the presence of disulfide bridges that give the enzyme stability and are often important for their catalytic activity [14]. The *Bacillus subtilis* lipase (Lip A) structure is presented in Figure 1.

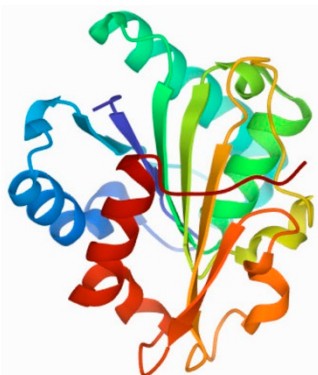

**Figure 1.** *Bacillus subtilis* lipase (Lip A) structure [15].

The classification of lipases is based on the sources of these enzymes and their specificity (Figure 2). Lipolytic microorganisms can be found in a variety of oil-contaminated habitats such as soil contaminated with oil, wastes of vegetable oils, dairy waste and deteriorated food [16]. Lipases are widely distributed in animals, plants and microorganisms, where their main role is to metabolize lipids [17]. Animal lipases are extracted, for the most part, from the pancreas of pigs and sheep, but they are also isolated from the liver, stomach and intestinal wall. In plants, lipases are located in seeds and vegetative organs [18]. However, most lipases that are studied and used industrially are obtained from microbial sources, such as bacteria, yeast and fungi, due to their unique properties, such as their superior stability, selectivity and substrate specificity position. Most microbial lipases are glycoproteins and they may be intracellular, extracellular or membrane bound [19]. Lipase selectivity is related to its preference to perform given reactions. The following three types of selectivity can be distinguished: substrateselectivity, regioselectivity and enantioselectivity. The regioselective lipases can be further divided based on their selective functionality, as follows: non-specific lipases, 1,3 specific lipases, and fatty acid-specific lipases. Substrate-specific lipases act selectively on a specific substrate, for example tri-, di- or monoglycerides in a reaction mixture facilitate the synthesis of the desired product. Substrates that can be acted upon by substrate-specific lipases include fatty acids and alcohols [20]. Regioselectivity is defined as the lipase reaction towards a given ester bond in the glycerol backbone of triglycerides, such as the primary or secondary ester bond. Regioselective lipases have the ability to steer the reaction in a favorable direction

as compared to other side reactions. This specific property of lipases is important to both the chemical and pharmaceutical industries, such as in the production of isomeric compounds, which show optimal function only under specific configurations. Non-specific lipases catalyze the hydrolysis of triacylglycerols into free fatty acids and glycerol. The 1,3-specific lipases catalyze the hydrolysis of triacylglycerols at the C1 and C3 positions, producing fatty acids, 1,2- or 2,3- diacylglycerols and 2-monoacylglycerols. Fattyacid-specificlipaseshydrolyzeesters, whichhavelong-chainfattyacids with doublebonds on C-9, such as the lipase from *Geotrichumcandidum* [21]. Enantioselectivity refers to the hydrolysis by lipases of one of the isomers of a racemate over the other from prochiral precursors. Additionally, this type of lipase can differentiate between enantiomers in a racemic mixture [22].

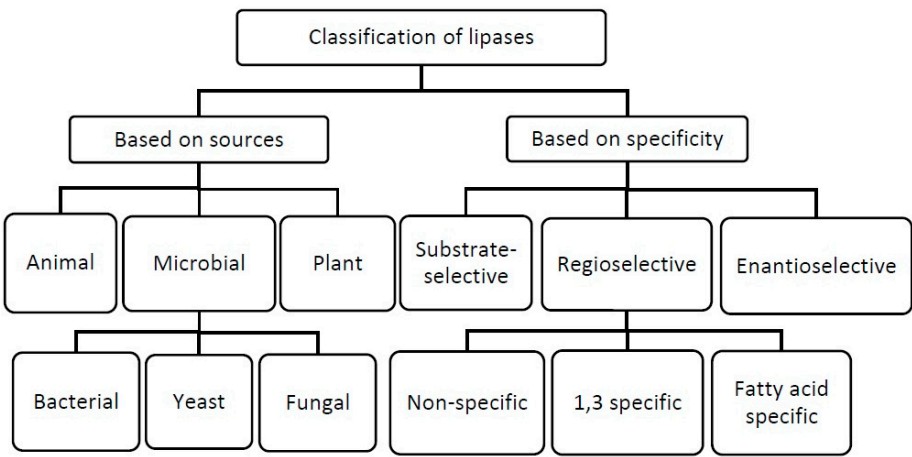

**Figure 2.** Classification of lipases.

## 2. Functions and Applications of Lipases

Lipases are ubiquitous enzymes with an important physiological role and industrial potential, because they catalyze hydrolysis and synthesis reactions. Synthesis reactions can be classified as esterification, transesterification, alcoholysis, acidolysis, interesterification, aminolysis. The alcoholysis, acidolysis, interesterification and aminolysis reactions as a group are considered to be transesterification reactions. In all cases, the reaction is carried out at the interface of a biphasic system reaction, which contains an organic and hydrophobic phase. Moreover, the catalytic activity of lipases is reversible [8,14,20]. Selected types of reactions catalyzed by lipases are presented in Figure 3. Lipases catalyze the hydrolysis of the ester bond of fats, oils and esters with the release of free fatty acids, diglycerides, monoglycerides and glycerol or alcohol in an aqueous system (Figure 3a). Lipases have the property of reversing hydrolytic reactions under the conditions of a micro-aqueous system, which also leads to esterification and transesterification. Organic solvents, such as methanol, ethanol, dioxane and hexane, allow fat or oil to be solubilized and converted from a two-phase system into a one-phase system. The advantage of using organic solvents in lipase-catalyzed esterification and transesterification reactions is that the water content can be controlled [23]. The optimum water content for esterification and transesterification by different lipases ranges from 0.04% to 11% (*w/v*). Moreover, most cases require a water content of less than 1% for effective synthesis reactions [24]. Esterification is a double-displacement reaction between alcohols and carboxylic acids, resulting in esters and water (Figure 3b) [25]. Esterification catalyzed by lipase may be an alternative to organic ester production because water is the only byproduct of the reaction. Additionally, lipase-catalyzed ester synthesis does not involve the use of hazardous solvents, and water may be removed during the process of esterification, making the conversion more effective. Transesterification is categorized into four subclasses according to the chemical species in which the organic group of an ester is replaced by an alcohol, acid or ester. Alcoholysis is the reaction between an ester and an alcohol (Figure 3c) [26], while acidolysis is the

reaction between an ester and an acid (Figure 3d) [27]. Interesterification is the reaction between two different esters, where the alcohol and acid moiety is swapped (Figure 3e) [28]. In aminolysis, an ester is reacted with an amine, generating an amide and an alcohol (Figure 3f) [8].

**a) Hydrolysis**
$R_1COOR_2 + H_2O \rightarrow R_1COOH + R_2OH$
**b) Esterification**
$R_1COOH + R_2OH \rightarrow R_1COOR_2 + H_2O$
**c) Alcoholysis**
$R_1COOR2 + R_3OH \rightarrow R_1COOR_3 + R_2OH$
**d) Acidolysis**
$R_1COOR_2 + R_3COOH \rightarrow R_3COOR_2 + R_1COOH$
**e) Interesterification**
$R_1COOR_2 + R_3COOR_4 \rightarrow R_1COOR_4 + R_3COOR_2$
**f) Aminolysis**
$R_1COOR_2 + R_3NH_2 \rightarrow R_1CONHR_3 + R_2OH$

**Figure 3.** Types of reactions catalyzed by lipases: (**a**) hydrolysis, (**b**) esterification, (**c**) alcoholysis, (**d**) acidolysis, (**e**) interesterification, (**f**) aminolysis [29].

The capability of catalyzing various reactions makes lipases very useful biocatalysts for the following applications: flavor modification in the food industry [30], detergent additive—for the hydrolysis of fats [31], pharmaceutical products—for the digestion of oil and fats in foods [19], leather processing—for the removal of lipids from animal skins [32], textile production—as a wax-removing agent [33], cosmetic production—for the removal of lipids [20], paper processing—for the hydrolysis of triglycerides in resins, and other synthetic chemistry applications, for example, the production of biopolymers and biodiesel [34,35] (Figure 4). The global microbial lipase market was valued at USD 340 million in 2020. It is projected to reach USD 494.7 million by the end of 2026, growing at a CAGR of 5.4% during 2021–2026 [36]. Lipase catalyzes the hydrolysis of milkfat, the synthesis of baked foods, wines, emulsifiers and supplements, and contributes to flavor enhancement in cheeses, creams and other milk products [37]. The use of anencapsulatedenzymecontains a lipase (Palatase M from Novozymes), and a proteasecan be used to obtain a full-flavoredcheesewithout a bittertaste [38]. Lipases are useful in the synthesis of structured lipids (SLs) through esterification and interesterification reactions. Structural lipids are lipids that have been modified from their natural biosynthetic form in order to conform to certain desired nutritional, physicochemical or textural properties for various applications in the food industry [39]. Sánchez et al. [40] reported that *Bacillus subtilis* plays a major role in bread making. Due to the excellent capability of lipases to catalyze various reactions, especially for the resolution of racemic mixtures, or the ability to catalyze reactions even in biphasic systems, they are useful in the medical and pharmaceutical industry. Sikora et al. [41] synthesized profens (2-aryl propionic acids) through enantioselective esterification, catalyzed by lipases from *Candida rugosa* and *Candida antarctica*. Moreover, bacterial lipases can replace pancreatic lipases, which are used to treat pancreatitis and cystic fibrosis [42]. Lipases are used in combination with amylases and proteases to improve the efficacy of detergents. Lipase adsorbs onto the fabric surface and constitutes a stable complex with the fabric, which catalyzes the breakdown of chemical bonds upon the addition of water [43]. A lipase from *Pseudomonas fluorescens* has exhibited the ability to remove oil stains from a woolen cloth [44]. Lipases have found applications during the stages of soaking, bating and degreasing in leather processing. In this case, expensive and hazardous solvents can be replaced by lipases, which have the ability to breakdown lipids. Degreasing suede clothing leathers obtained from wooled sheep skins using lipase originating from *Rhizopus nodosus* demonstrated a better quality degreasing process when

compared with the application of a conventional solvent treatment [45]. Bacterial lipases may be used to synthesize agrochemicals and to obtain biodiesel. Transesterification using lipases from *Pseudomonas* allows for insecticides, fungicides and herbicides such as (S)-indanofan to be obtained, this is used to control wild grass and weeds [46]. Moreover, optically active forms of pesticides were synthesized using lipases from *Bacillus subtilis* for the resolution of racemic mixtures of carboxylic esters or alcohol [47]. Biodiesel is defined as mono-alkyl esters of long-chain fatty acids, which are usually methyl esters of fatty acids and are commonly denoted as fatty acid methyl esters (FAME), which are used as an alternative fuel that is non-toxic, non-inflammable, biodegradable and renewable. Biodiesel can be produced from vegetable oils, such as sunflower oils, rapeseed oils, animal fat, palm oil, rice bran oil and soybean oil. Microbial lipases catalyze the transesterification reaction between lipids and alcohols to produce an ester and glycerol. Due to the transesterification reaction, the production of biodiesel requires a micro-aqueous system. Many microbial lipases have the ability to produce biodiesel through transesterification using agro-industrial residues, for example, *Bacillus subtilis* produces biodiesel from waste cooking oil, and *Burkholderiacepacia*, *Enterobacter aerogenes* or *Chromobacteriumviscosum* produces biodiesel from jatropha oil [48]. The different commercial lipases used in the industry are summarized in Table 1.

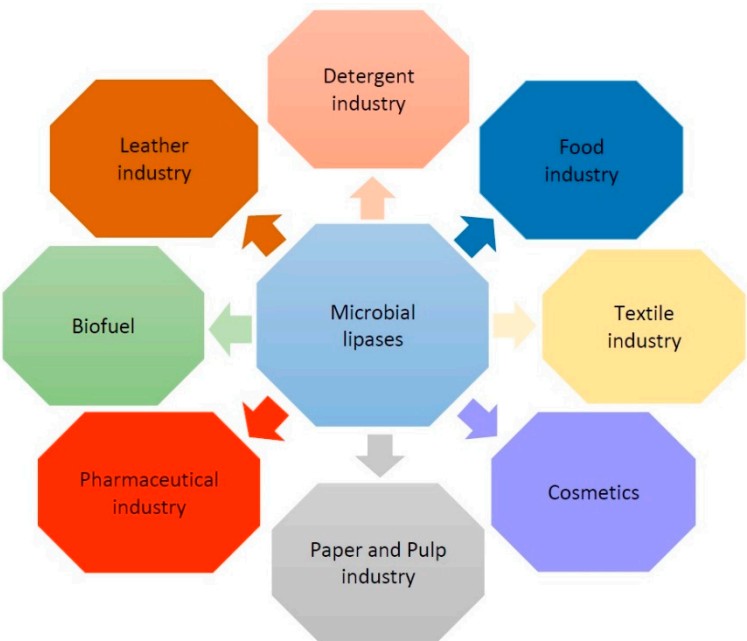

**Figure 4.** Application of microbial lipases in different branches of industry.

**Table 1.** Available commercial lipases [49].

| Commercial Lipases | Catalogue Number | Form | Specific Activity | Definition of the Unit of Specific Activity | Price |
|---|---|---|---|---|---|
| Lipase from *Aspergillus oryzae* | 62285 | lyophilized powder | ~50 U/mg | 1 U corresponds to the amount of enzyme that liberates 1 μmol oleic acid per minute at pH 8.0 and 40 °C. | 100 mg/158 USD |
| Lipase from *Pseudomonas* sp. | L9518 | lyophilized powder | ≥15 units/mg solid | One unit will produce 1.0μmol of glycerol from a triglyceride per minute at pH7.0 at 37 °C in the presence of bovine serum albumin. | 500 units/219 USD |
| Lipase from *Candida rugosa* | 62302 | lyophilized solid | 15–25 U/mg | 1 U corresponds to the amount of enzyme that liberates 1 μmol oleic acid per minute at pH 8.0 and 40 °C. | 100 mg/207 USD |

**Table 1.** *Cont.*

| Commercial Lipases | Catalogue Number | Form | Specific Activity | Definition of the Unit of Specific Activity | Price |
|---|---|---|---|---|---|
| Lipase from *Rhizopus oryzae* | 62305 | powder (fine) | ~10 U/mg | 1 U corresponds to the amount of enzyme that liberates 1 μmol of butyric acid per minute at pH 8.0 and 40 °C, and 5000 U as described above are equivalent to ~1 U using triolein at pH 8.0 and 40 °C. | 1 g/109 USD |
| Lipase from *Aspergillus niger* | 62301 | powder (fine) | ~200 U/g | 1 U corresponds to the amount of enzyme that hydrolyses 1 μmol acetic acid per minute at pH 7.4 and 40 °C, and 50 U as described above are equivalent to ~1 U using triolein substrate, at pH 8.0 and 40 °C. | 100 mg/23.90 USD |
| Lipase from *Pseudomonas cepacia* | 62309 | powder | ≥30 U/mg | 1 U corresponds to the amount of enzyme that liberates 1 μmol oleic acid per minute at pH 8.0 and 40 °C. | 100 mg/89.70 USD |
| Lipase from *Candida* sp. | L3170 | Aqueous solution | ≥5000 LU/g | Not defined. | 50 mL/77.20 USD |
| Lipase from *Mucor miehei* | 62298 | powder | ~1 U/mg | 1 U corresponds to the amount of enzyme that liberates 1 μmol oleic acid per minute at pH 8.0 and 40 °C. | 100 mg/86.50 USD |
| Lipase from *Mucor javanicus* | L8906 | lyophilized powder | ≥300 U/mg | One unit will hydrolyze 1.0 microequivalent of fatty acid from a triglyceride in 1 h at pH 7.7 at 37 °C using olive oil (30 min incubation). | 1 g/271 USD |
| Lipase from *Candida antarctica* | 02569 | lyophilized powder | ~0.3 U/mg | 1 U corresponds to the amount of enzyme that liberates 1 μmol oleic acid per minute at pH 8.0 and 40 °C. | 100 mg/118 USD |
| Amano Lipase PS from *Burkholderiacepacia* | 534641 | powder | ≥30,000 U/g, pH 7.0, 50 °C (optimum pH and temperature) | Not defined. | 10 g/60.10 USD |
| Amano Lipase from *Pseudomonas fluorescens* | 534730 | powder | ≥20,000 U/g, pH 8.0, 55 °C (optimum pH and temperature) | Not defined. | 10 g/51.20 USD |

## 3. Fermentation Techniques

Microbial processes occurring with or without air are defined as fermentation. Fermentation has been widely used for the production of various substances, for example, enzymes such as lipases, which are beneficial to the industry and individuals. Different microorganisms have been employed for lipase production, varying from a prokaryotic system involving both Gram positive and Gram negative bacteria, to a eukaryotic system such as yeast and filamentous fungi. Microorganisms could be cultivated on solid as well as on liquid media. Submerged fermentation (SmF) and solid-state fermentation (SSF) are the two important fermentation technologies for the production of lipases by microorganisms such as bacteria, yeast or filamentous fungi under carefully controlled conditions due to the ease of multiplication and handling. The great advantage of both techniques is that they can use renewable agro-industrial residues and wastes. Most researchers use the frying oil from olives and sunflowers [50] or olive mill wastewater [51] for submerged fermentation, and wheat bran [52], soybean bran [53] or oil cakes [54] for solid-state fermentation. These technologies offer several benefits, but they are different in several respects [55].

Submerged fermentation is defined as the fermentation process occurring in the presence of an excess of water. Submerged fermentation presents the opportunity to cultivate microorganisms such as bacteria, yeast or filamentous fungi in closed vessels containing a liquid nutrient broth. Closed vessels are equipped with a suitable spargerthat introduces oxygen to the fermentation medium, and a stirrer that mixes biomass, suspended particles and gas in the fermenters. The cultivated microorganisms digest the nutrients and grow, they also release the desired enzymes into the medium. When the selected strain produces the enzyme extracellularly, it makes purification and recovery easier than when it is produced intracellularly. Most industries employ submerged fermentation for enzyme production due to the ease of handling on a large-scale when compared to solid-state fermentation. Different methods of cultivating the microorganisms in the submerged fermentation were applied, such as batch culture, fed-batch culture and continuous culture. In the batch culture, the microorganisms are cultivated in a specific volume of the medium located in a closed vessel. In the fed-batch culture, the components of the medium are added to the batch culture. In the continuous culture, the medium is added to the batch culture at the exponential phase of microbial growth and withdrawal with the medium containing the obtained product. Due to the use of complex equipment, submerged fermentation allows for the online control of several parameters such as temperature, pH, stirring speed, gas supply, optical density, and the analysis of the oxygen and carbon dioxide concentrations of the exit gas [56–58].

Solid-state fermentation may be understood as the fermentation process occurring in the absence or near-absence of water. Solid-state fermentation presents the opportunity to cultivate microorganisms such as bacteria, yeast or filamentous fungi on moist solid supports or on inert carriers, or else on insoluble substrates that could be used as an energy and carbon source [57]. On a laboratory scale, solid-state fermentation offers several benefits, such as the cultivation of microorganisms specialized for water-insoluble substrates or the mixed cultivation of various microorganisms, the lower demand on microbial sterility due to the absence of water, higher fermentation productivity and a higher concentration of products. On the other hand, enzyme production on a large-scale using solid-state fermentation generates severe engineering problems due to the rise in temperature, pH or substrate and moisture gradients. However, solid-state fermentation offers more economic and environmental benefits than submerged fermentation [59]. On a laboratory scale, solid-state fermentation is carried out using standard laboratory glassware as bioreactors, such as Erlenmeyer flasks or Petri dishes. Different methods of cultivating the microorganisms in solid-state fermentation on a large-scale were applied, such as the tray, rotating drum and continuous screw bioreactors. Most solid-state fermentation bioreactors operate in batch mode. Tray bioreactors are made from wood, plastic or metal. The trays are filled with the moistened and inoculated solid substrate to a bed height from 5 to 15 cm, and moved to a room with controlled temperature and humidity [60]. Rotating drum bioreactors consist of a horizontal drum, which may have baffles. Rotating drum bioreactors are filled with a solid substrate from 10% to 40% of reactor volume. Rotating drum bioreactors allow for the control of several parameters, such as air supply, rotation speed and frequency of mixing [61]. Continuous screw reactors are based on screw conveyors that can move filled solid substrate with almost zero mixing in the direction of flow. Inoculation may be achieved by recycling a part of the fermented product. Additionally, this type of inoculation avoids the need for a separate process of inoculum production. Rotating drum and continuous screw bioreactors can be operated in either batch or continuous mode [62]. The differences between submerged fermentation and solid-state fermentation are presented in Table 2.

**Table 2.** Comparison between solid-state fermentation (SSF) or submerged fermentation (SmF) [56].

| Factor | Solid-State Fermentation (SSF) | Submerged Fermentation (SmF) |
| --- | --- | --- |
| Type of process | Batch | Batch or continuous |
| Substrates | Insoluble | Soluble |
| Water consumption in the fermentation process | Low | High |
| Amount of inoculum | Low (usually in liquid form added to medium) | High (usually added on surface of medium) |
| Contamination | High risk of contamination | Low risk of contamination |
| Energy consumption | Low | High |
| pH control | Difficult to control the pH parameter | Easy to control the pH parameter |
| Temperature maintenance | Difficult to control the temperature | Easy to control the temperature |
| Aeration | Better oxygen circulation | Aeration using sparger |
| Agitation | Low rpm (static conditions) | High rpm (homogenisation) |
| Equipment | Low costs and simple equipment | High costs and complex equipment |
| Scale-up | Difficult to scale-up because of engineering and equipment | Engineering and equipment are available |
| Amount of effluents | Low | High |

## 4. Methods of Lipolytic Activity Determination

In order to evaluate the effectiveness of lipase production and the lipolytic activity of the microbes used in this process, various methods were developed and described in the literature, such as titrimetry, spectroscopy, photometry, fluorimetry, chromatography, tensiometry, turbidimetry, conductometry, immunochemistry, and microscopy [63]. Turbidimetric, conductimetric, tensiometric and microscopic methods allow for the detection of changes in the biophysical properties of substrates during lipolysis. Moreover, these methods are difficult to establish and require expensive equipment [63]. Lipase-producing strains can be determined qualitatively through the use of agar plates or gel-diffusion assay, and quantitatively using titrimetry, colorimetric assays and chromatographic procedures. An important method for the selection of microbial lipases is the agar plates assay, wherein lipolysis is observed directly through changes in the appearance of the substrate. Different researchers mainly used agar plates supplemented with olive oil and rhodamine B, Tributyrin or Tween 80. The lipolytic activity in olive oil using rhodamine B agar plates includes triglyceride hydrolysis in the olive oil and was detected through the appearance of orange fluorescence halos, which are formed between the cationic rhodamine B and free fatty acids around the colonies under UV light (Figure 5a). Lipolytic enzymesproduced by the microorganisms hydrolyze the substrates, resulting in the transparency of the medium around the colonies grown on tributyrin agar plates (Figure 5b). The hydrolysis of Tween 80 releases fatty acids, which bind with the calcium in the medium to form insoluble crystals around the point of inoculation (Figure 5c) [64].

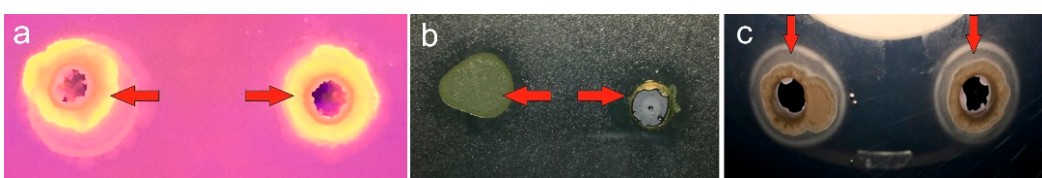

**Figure 5.** Lipase-producing strains: (**a**) *Yarrowialipolytica* ATCC 18942 in olive oil using a rhodamine B agar plate, (**b**) *Burkholderiacepacia* DSM 7288 on a tributyrin agar plate, (**c**) *Moesziomycesantarcticus ATCC 28323* on a Tween 80 agar plate.

Titrimetry is one of the oldest and most widely used quantitative assays on account of its simplicity, accuracy and reproducibility. Triolein or olive oil, which contains about 70% triolein, is a generally used and widely accepted substrate for the quantification of released fatty acids, due to the fact that lipases have a higher rate of hydrolysis on long-chain triacylglycerols. One unit of lipase activity may be defined as the amount of enzyme that catalyzes the release of 1 μmol of fatty acid from olive oil/min per ml under standard assay conditions. Moreover, titrimetry is extremely time consuming and laborious [65]. Colorimetric methods are based on the hydrolysis of ester substrates by lipolytic enzymes, and the release products can be detected using spectrophotometry. Usually the lipolytic activity method is based on the quantification of p-nitrophenol (p-NP) through UV–VIS radiation, produced during the hydrolysis of p-nitrophenyl dodecanoate (C12), myristate (C14), palmitate (C16) and stearate (C18), indicating that they could be characterized as potential lipases because of their chain length. Lipolytic activity was determined by measuring the release of p-nitrophenol through monitoring the UV/VIS levels. One unit of enzyme activity was defined as the amount of enzyme required to release 1 μmol of p-NP per min under assay conditions. Moreover, the units of lipolytic activity per microgram of extracellular protein expressed the specific activity of the lipases. Colorimetric and fluorimetric assays are simpler and more rapid than titrimetry, but the substrates are relatively expensive [66]. The products of lipolysis can also be easily identified using HPLC. Gulomova et al. [67] used HPLC to analyze the reaction products and showed that the hydrolysis of triolein by *Penicillium* sp. leads to the accumulation of diolein and monoolein, indicating the reduced rate of the hydrolysis of both diglyceride and monoglyceride.Kulkarni and Gadre [68] used the assay method for lipase activity determination by gas chromatography. Tributyrin is hydrolyzed by AmanoPS (*Pseudomonas fluorescens*) lipase to produce butyric acid. The enzyme activity was measured in terms of the free butyric acid produced, which is directly estimated using gas chromatography.

## 5. Agro-Industrial Residues and Wastes for Lipase Production

Although commercially available lipases are widely used as biocatalysts, there are still many challenges concerning the production of microbial lipases, with the use of renewable sources as the main component of microbial growth medium such as straw, bran, oil cakes and industrial effluents [17]. Agro-industrial residues and wastes may be divided into two different types, such as agricultural wastes and industrial wastes (Figure 6). Agricultural wastes can be further divided into field residues, which refers to the harvesting of crops and process residues, which corresponds to a harvest that has been further processed into a different raw material. Agricultural wastes include stems, leaves, seed pods, husks, seeds, roots, bagasse and molasses. Industrial wastes mainly consist of the waste produced by the food processing industries, and includes peels, oil cakes and waste water [18]. The food and agricultural industry is expanding rapidly. The rapidly growing population and demand for food has resulted in the amount of agro-industrial residues and wastes increasing each year, and this will lead to serious environmental and economic problems. In response to this fact, one of the most important and urgent challenges is to create new uses for agro-industrial wastes and residues. Additionally, the use of agro-industrial residues and wastes as raw materials can help to reduce production costs and also reduce the pollution load on the environment [69]. Moreover, overproduction, damage during harvesting, infections caused by microbes, insects and pests, postharvest handling, inadequate transport and storage, food processing, improper distribution, and wasteful consumer usage are the main reasons for food wastage [70]. Most residues and wastes generated by the food and agricultural industry are a source of carbohydrates, fat, protein, lignin and cellulose. These ingredients facilitate microbial growth such as bacteria, filamentous fungi and yeast [71]. The composition of selected agro-industrial residues and wastes, employed for the cultivation of microorganisms that can produce lipases, is presented in Table 3. Despite this fact, agro-industrial residues and waste, such as bran, straw, molasses, peels, seeds, pomace, waste water and oil cakes, can be used for the production of various

value-added products, for example, for the production of industrially important enzymes such as lipases [72], biofuels [73], antibiotics [74], mushrooms [75] and also various useful chemicals [76].

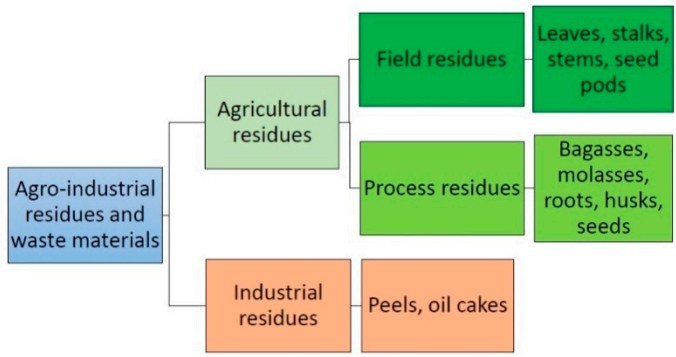

**Figure 6.** Different types of agro-industrial residues and waste materials.

**Table 3.** Composition of agro-industrial residues and wastes.

| Substrates | Dry Matter (%) | Crude Protein (%) | Crude Fat (%) | Crude Fiber (%) | Crude Ash (%) | Source |
|---|---|---|---|---|---|---|
| Rapeseed cake | 95.3 | 36.1 | 12.2 | 13.1 | 7.1 | [77,78] |
| Mustard cake | 91.4 | 30.2 | 6 | 6 | 6.7 | [79,80] |
| Sunflower cake | 93.8 | 33.7 | 8 | 18.5 | 6 | |
| Soy Bean cake | 93.2 | 44 | 9 | 6 | 6 | [81–83] |
| Corn germ cake | 87.8 | 18.9 | 2.5 | 7.9 | 5.7 | |
| Wheat bran | n. d. | 18 | 5.3 | 9.4 | 5 | |
| Rice hulls | n. d. | 3 | 0.8 | 20 | 22.3 | |
| Mustard meal | n. d. | 34 | n. d. | 1.8 | 7.5 | [84–88] |
| Coconut meal | n. d. | 21.3 | n. d. | 2 | 6 | |
| Almond meal | n. d. | 22 | n. d. | 4 | 7 | |
| Grape pomace | 52.2 | 14.1 | n. d. | 27.5 | 7.8 | |
| Almond Hull | 31 | 3.2 | n. d. | 13 | 8.4 | [89–92] |
| Pomegranate peel | 31.3 | 2.6 | n. d. | 13.5 | 5.7 | |
| Dried apple pomace | 89.56 | 5.5 | 4.8 | 17.99 | 3.4 | |
| Palm oil mill effluent | n. d. | 12.7 | 10.2 | n. d. | 14.9 | [93,94] |

n. d.—not determined.

## 6. Oil Cakes

In the European Union, oilseed production for 2019/2020 amounted to 15.36 million tons for rapeseed, 10.07 million tons for sunflower seeds and 2.69 million tons for soya beans, respectively [95]. Oil cakes or oil meals are solid residues that are produced by a simple oil pressing system or extraction technique from rapeseed, sunflower or soya bean. Oil cakes and oil meals have a different chemical composition depending on the plant source, extraction method, growing conditions and quality of the seeds [96]. There are the following two types of oil cakes: edible and non-edible. The main advantage of edible oil cakes is that they contain a high nutritional value, with an especially high protein content ranging from 15 to 50%, which allows for feed applications for fish, poultry [97], pigs, and ruminants [98]. Oil cakes are an excellent source of amino acids, for example, methionine, tryptophan, cysteine, lysine or threonine. The amino acid composition of oil cakes depends on their type. Sunflower oil cakes can be divided into the following three main components: the lignocellulosic fraction (23.2–25.3% dry weight), the proteinaceous fraction (55.4–57.6% dry weight) and the soluble fraction (17.1–21.4% dry weight), and have a high methionine content but a low lysine and threonine content. Moreover, a large

proportion of the proteins (about 80–90%) are linked to the lignocellulosic fraction [99]. Non-edible oil cakes, including neem (*Azadirachtaindica*), castor (*Ricinus communis*), karanja (*Pongamiapinnata*), jatropha (*Jatropha curcas*) and mahua (*Madhucaindica*), are a rich source of micro and macronutrients such as N, P and K. Due to their N, P and K contents, they may be utilized as plant nutrients. Despite their nutritious quality, some oil cakes have their feedstock potential limited by toxic compounds such as curcin from jatropha, and karanjin, pongamol and glabrin from karanja. On the other hand, non-edible oil cakes can be used to control pests in the soil and simultaneously they may be applied for the purposes of supplying nutrients in the form of organic fertilizers [100]. Because of their high nutrient value, oil cakes have been reported to be good substrates for microbial enzyme production. The earliest reports about lipase production were published by Ramakrishnan and Banerjee [101]. *Penicillium chrysogenum* S1 demonstrated lipase activity on sesame cake medium containing 10% sesame oil. Oliviera et al. [72] conducted an extensive study to optimize the lipase production from different oil cakes produced in Brazil, through solid-state fermentation using *Aspergillus ibericus* MUM 03.49. Through SSF optimization, using palm kernel oil cake mixed with sesame oil cake in a ratio of 0.45 g of palm kernel oil cake per g of total substrate at a 57% moisture content, the production of 460 U/g of lipase was obtained after 6 days of fermentation. Parihar [80] used a ground nut linseed and mustard oil cake to produce lipase through the activity of *Pseudomonas aeruginosa* LB-2. The optimum fermentation condition was carried out for 50 h at 30 °C and at a moisture content of 50%. Moreover, the addition of olive oil to the fermentation medium increased the production of lipase up to 19.2 U/g substrate. In another study, solid-state fermentation was carried out by Shukla et al. [102] at 30 °C for 7 days for lipase production from *Rhizopus oryzae* KG-10, using different low cost available oil cakes such as jatropha oil cake, flaxseed oil cake, mustard oil cake and groundnut oil cake. Among the four substrates used, the crude enzyme extracted from the mustard oil cake medium showed the highest activity of 170 IU. The efficacy of sesame oil cake as a substrate for lipase production, by the marine *Bacillus sonorensis* strain in submerged fermentation, has been studied and reported by Nerurkar et al. [103]. Different physical and chemical parameters, such as pH, temperature, substrate concentration and incubation time, were optimized. The lipase activity was found to reach a maximum level when the pH of the medium containing 4% of the mustard oil cake was 5.0 at 40 °C, after 48 h of submerged cultivation. Boratyński et al. [78] investigated rapeseed oil cakes as a microbial medium for lipase biosynthesis. Twenty-six different filamentous fungi were screened using solid-state fermentation. Additionally, various process parameters were optimized to maximize lipase production, such as the carbon and nitrogen source, metal ions, temperature, moisture content, initial pH, and inoculum size. During the fermentation process, the production of lipase showed its highest activity at 247 U/mg in solid-state cultures on rapeseed cake supplemented by lactose and calcium chloride, alkalinized to pH 8, hydrated to 80%, and inoculated with $1.2 \times 10^6$ spores/mL. Another study, conducted by Sahoo et al. [104], reveals the application of biosurfactants in lipase production in solid-state fermentation using olive oil cake obtained from the local manufacturing units as a substrate and with the thermophilic bacteria *Bacillus licheniformis* JQ991000. The optimum parameters were determined, with the maximum lipase activity calculated to be 61.5 U/g at a pH of 8.2, a temperature of 50.8 °C, a moisture content of 55.7% and a biosurfactant content of 1.693 mg. Rajendran and Thangavelu [105] used *Candida rugosa* NCIM 3462 for the production of lipase in the solid-state fermentation of agro-industrial wastes such as sesame oil cake, groundnut oil cake and coconut oil cake. The maximum lipase activity of 22.40 U/g substrate was obtained using the sesame oil cake. The optimized temperature and substrate-to-moisture ratio were found to be 32.3 °C and 1:3.23 g/mL, respectively. Priya et al. [106] conducted an extensive study to screen actinomycete strains from sediments. Out of the 34 strains tested, 26 strains exhibited lipase activity. The *Streptomyces indiaensis* strain showed a maximum lipase activity of 220.8 IU/mL at pH 9 and at a temperature of 55 °C. The highest lipase activity was observed on the 4th day with 80% moisture content and solid-state fermentation

using coconut oil cake with both of the following inducers: soy meal and wheat bran. The efficiency of various substrates such as groundnut oil cake, neem oil cake, mustard oil cake and linseed oil cake, as a substrate for lipase production by *Bacillus subtilis* in solid-state fermentation, has been assessed by Chatuverdi et al. [107]. Different process parameters were optimized, such as temperature, pH, various substrates and incubation time. The maximum lipase activity of 4.5 U/g substrate was observed with the groundnut oil cake after 48 h of solid-state fermentation at pH 8 with a 70% moisture content. El Aal et al. [108] used the *Aspergillus niger* NRRL-599 strain for lipase production, using different agro-industrial wastes including wheat bran, wheat germ cake oil, jojoba cake oil, almond cake oil, and olive oil as substrates under solid-state fermentation. A maximum lipase activity of 200 U/mL and specific enzyme activity of 357.1 U/mg were obtained in the presence of 5% $w/v$ olive oil after 14 days of fermentation at 30 °C. The different microorganisms, oil cakes in combination with other substrates used in the fermentation process, type of fermentation and maximum lipase activity reported in the literature for the production of lipases are summarized in Table 4.

**Table 4.** Production of lipase using oil cakes.

| Substrates | Microorganism | Type of Fermentation | Maximum Lipase Activity | Source |
|---|---|---|---|---|
| Andiroba oil cake | *Aspergillus ibericus*MUM 03.49 | SSF | 1 U/g | [72] |
| Cupuassu oil cake | *Aspergillus ibericus*MUM 03.49 | SSF | 11 U/g | [72] |
| Canola oil cake | *Aspergillus ibericus*MUM 03.49 | SSF | 47 U/g | [72] |
| Macauba oil cake | *Aspergillus ibericus*MUM 03.49 | SSF | 1 U/g | [72] |
| Palm kernel oil cake | *Aspergillus ibericus*MUM 03.49 | SSF | 127 U/g | [72] |
| Crambe oil cake | *Aspergillus ibericus*MUM 03.49 | SSF | 44 U/g | [72] |
| Green coffee oil cake | *Aspergillus ibericus*MUM 03.49 | SSF | 4 U/g | [72] |
| Jatropha oil cake | *Rhizopus oryzae*KG-10 | SSF | 80 IU | [102] |
| Teesi oil cake | *Rhizopus oryzae*KG-10 | SSF | 60 IU | [102] |
| Mustard oil cake | *Rhizopus oryzae*KG-10 | SSF | 170 IU | [102] |
| | *Bacillus subtilis MTCC 6808* | SSF | 1.68 U/g | [107] |
| | *Pseudomonas aeruginosa* LB-2 | SSF | 19.2 U/g | [80] |
| Groundnut oil cake | *Rhizopus oryzae*KG-10 | SSF | 60 IU | [102] |
| | *Bacillus subtilis MTCC 6808* | SSF | 4.5 U/g | [107] |
| Sesame oilcake | *Bacillus sonorensis* | SmF | 41.35 U/mL/min | [103] |
| | *Aspergillus ibericus*MUM 03.49 | SSF | 78 U/g | [72] |
| | *Candida rugosa NCIM 3462* | SSF | 22.40 U/g | [105] |
| Rapeseed oil cake | *Aspergillus* sp. AM31; *Aspergillus candidus*AM386; *Aspergillus glaucus*AM211; *Aspergillus nidulans*AM243; *Aspergillus ochraceus* AM456; *Aspergillus wenthi*AM413; *Botrytis cinerea*AM235; *Fusarium avenaceum*AM11; *Fusarium equiseti* AM15; *Fusarium oxysporum*AM13; *Fusarium oxysporum*AM21; *Fusarium semitectum*AM20; *Fusarium solani*AM203; *Fusarium tricinctum*AM16; *Mucor spinosus*AM398; *Papulariarosea*AM17; *Penicillumcamembertii* AM83; *Penicillium chrysogenum*AM112; *Penicillium frequentans*AM351; *Penicillium notatum* AR904; *Penicillium Thomii*AM91; *Penicillium vermiculatum*AM30; *Poria placenta* AM38; *Sclerophomapythiophila*AR55; *Spicoria divaricate* AM423; *Syncephalastrumracemosum*AM105 | SSF | 247 U/mg | [78] |

| Substrates | Microorganism | Type of Fermentation | Maximum Lipase Activity | Source |
|---|---|---|---|---|
| Olive oil cake | *Bacillus licheniformis* | SSF | 62.3 IU/g | [104] |
| Coconut oil cake | *Bacillus subtilis MTCC 6808* | SSF | 2.35 U/g | [107] |
| | *Streptomyces indiaensis* | SSF | 220.8 IU/mL | [106] |
| Neem oil cake | *Bacillus subtilis MTCC 6808* | SSF | 1.33 U/g | [107] |
| Linseed oil cake | *Bacillus subtilis MTCC 6808* | SSF | 1.24 U/g | [107] |
| Wheat germ oil cake | *Aspergillus niger NRRL-599* | SSF | 80 U/mL | [108] |
| Jojoba oil cake | *Aspergillus niger NRRL-599* | SSF | 60 U/mL | [108] |
| Almond oil cake | *Aspergillus niger NRRL-599* | SSF | 90 U/mL | [108] |

## 7. Fibrous Residues

Agricultural and agro-industrial activities produce a large number of residues, such as sugarcane bagasse, citrus bagasse, fruit peel, corn straw and corncobs. Agriculture is the source of two types of residues, such as fibrous residues and brans. Fibrous residues can be further divided into the following two types: those with a high degree of digestibility and those with a low degree of digestibility. These fibrous residues have different nutritional values. Highly digestible fibrous residues include citrus pulp, corn gluten bran, soy husk and brewing residues (barley). The less digestible fibrous residues include sugarcane bagasse, cereal, corn, straws, husks and the harvest remnants of forage grass seeds. Additionally, the brans include soy, cotton, rice and peanut [109]. Moreover, both the highly digestible and less digestible fibrous residues and brans can serve as a physical support, and as a source of carbon and nutrients to sustain microbial growth. One of the main fractions of the plant cell wall are fibers, which mainly consist of carbohydrates. The primary components of fiber are cellulose, hemicellulose and lignin. Cellulose and hemicellulose represent the largest fraction of agricultural residues, for example, sugarcane bagasse and straw from wheat. Cellulose consists of linear chains of glucose, which are linked together by β-1,4 glycosidic bonds that can be digested by enzymes of microbial origin. Hemicellulose is associated with lignin, which has a negative impact on fiber digestion. Nevertheless, there are many research areas that involve the use of various fibrous residues, either alone or in combination with other substrates, for nutrient augmentation [110].

An interesting study conducted by Amin and Bhatti [111] deployed a new potential fungal strain for lipase production using agro-industrial wastes as a substrate, such as rice bran, wheat bran, canola seed oil cake, sunflower hulls and peanut shells, under solid-state fermentation. Among the different residues, canola seed oil cake showed the highest lipase activity with 521 units/gram dry substrate (U/gds); this result was obtained after 48 h of fermentation in a medium containing 10 g of canola seed oil cake supplemented with 2% olive oil and with a 50% moisture content at an initial pH value of 4.0 at 30 °C. Colla et al. [88] studied the correlation between the production of lipases and biosurfactants through submerged and solid-state fermentation using *Aspergillus* spp. The culture medium for submerged fermentation was prepared with 10% (*w/v*) wheat bran, and the medium for solid-state fermentation was prepared with 85.7% (*w/w*) soybean meal and 14.3% (*w/w*) rice husk. The highest lipase activity was observed on the 4th day of submerged and solid-state fermentation. Additionally, the results obtained in this study show the simultaneous production of lipases and biosurfactants in a single bioprocess. Mala et al. [112] conducted an extensive study to optimize the solid-state fermentation process using *Aspergillus niger* MTCC 2594 and agro-industrial wastes as a substrate, such as wheat bran and gingelly oil cake. The lipase production showed the highest activity of 384.3 U/g of dry substrate containing wheat bran and gingelly oil cake in the ratio of 3:1 *w/w* at 30 °C over a period of 72 h. Fleuri et al. [113] evaluated the feasibility of using 10 fungal strains fermented in wheat bran, soybean bran and sugarcane bagasse for lipase production. No fungal growth was observed using sugarcane bagasse, but the

supplementation of the culture medium with wheat bran and soybean bran produced lipase production. The strains studied achieved maximum lipase activity with 25% sugarcane bagasse combined with 75% wheat bran or soybean bran at 40% moisture content. The optimum fermentation time for lipase production was 96 h. The effectiveness of wheat bran in the fermentation medium, as a carbon source and inducer for the production of lipase, was evaluated by Santos et al. [114]. *Aspergillus niger* 11T51A14 was inoculated onto the medium containing nitrogen (ammonium sulphate), sunflower soapstock and wheat bran. Additionally, the nitrogen concentration (0.32–0.88% $w/w$) and the volume of liquid (60.9–89.1 mL) in the solid-state fermentation was tested. The maximum enzyme activity found was 153.4 U/g, with a nitrogen concentration of 0.6% in an 89.1 mL volume of liquid at 32 °C for 72 h. The optimal conditions for the production of lipase in solid-state fermentation by *Rhizopus arrhizus* was studied by Dobrev et al. [115]. The maximum lipase activity was achieved when the strain was grown at 30 °C on wheat bran as a solid substrate. The addition of glucose at a concentration of 1% ($w/w$) as an additional carbon source, and tryptone at a concentration of 5% ($w/w$) as a source of organic nitrogen, stimulated lipase biosynthesis to a significant extent. The optimum moisture content value of the medium was determined to be 66% and the optimum tryptone concentration was 5% ($w/w$). As a result of the optimization, a value of 418.88 U/g lipase activity was achieved. Another study, conducted by Rekha et al. [116], concerned the screening of the microorganisms *Candida rugosa* NCIM 3467 and *Penicillumcitrinum* NCIM 765 with different agro-residues, such as rice bran, wheat bran, groundnut oil cake, coconut oil cake and sesame oil cake, to achieve the maximum production of lipases. Various process parameters were optimized to maximize lipase production, such as carbon and nitrogen supplementation, temperature, time, moisture content, initial pH, and inoculum age. The lipase production showed the highest activity of 63.35 U/mL in solid-state cultures at 5 days of fermentation, 32 °C, pH 6,a 15% inoculum level, 60% initial moisture content and supplemented with maltose (5% $w/w$) and peptone (3% $w/w$). Nema et al. [117] conducted an extensive study to optimize lipase production using a mixture of agro-industrial substrates, such as rice husk, cottonseed cake and red gram husk, in various combinations using solid-state fermentation. The maximum lipase activity was obtained at 28.19 U/g in the medium containing rice husk, cottonseed cake, and red gram husk combined in the ratio of 2:1:1 under the optimum cultivation conditions of a temperature of 40 °C, a moisture content of 75%, and pH 6.0. Pitol et al. [118] were able to achieve a maximum lipase enzyme activity of 264 U/g by adding wheat bran and sugarcane bagasse (50:50 $w/w$), supplemented only with urea using *Rhizopus microsporus* CPQBA 312-07 DRM as the enzyme producer. Awan et al. [119] used a mixture of almond meal and distilled water as a moistening agent as a substrate with a diluent ratio of 1:0.7, and 0.5% Tween 80 as a medium to produce lipase through the activity of *Rhizopus oligosporus* ISU-16 in solid-state fermentation. Fermentation was carried out for 48 h at 30 °C, which resulted in a maximum lipase yield of 71.65 U/g. The different microorganisms and fibrous residues in combination with other substrates in the fermentation process, the type of fermentation and the maximum lipase activity reported in the literature for the production of lipases are summarized in Table 5.

**Table 5.** Production of lipase using fibrous residues.

| Substrates | Microorganisms | Type of Fermentation | Maximum Lipase Activity | Source |
|---|---|---|---|---|
| Sunflower hulls | *Penicillium fellutanum* | SSF | 180 U/g | [111] |
| Rice bran | *Penicillium fellutanum* | SSF | 80 U/g | [111] |
| | *Rhizopus arrhizus* | SSF | 2.6 U/g | [115] |
| | *Candida rugosa* NCIM 3467 | SSF | 26 U/mL | [116] |
| | *Penicillumcitrinum* NCIM 765 | SSF | 31 U/mL | [116] |
| Peanut shells | *Penicillium fellutanum* | SSF | 150 U/g | [111] |

**Table 5.** *Cont.*

| Substrates | Microorganisms | Type of Fermentation | Maximum Lipase Activity | Source |
|---|---|---|---|---|
| Wheat bran | *Penicillium fellutanum* | SSF | 170 U/g | [111] |
| | *Aspergillus niger* | SmF | 4.52 U | [88] |
| | *Aspergillus niger* | SSF | 384.3 U/g | [113] |
| | *Penicillium* sp. | SSF | 6.6 U/mL | [113] |
| | *Aspergillus* sp. | SSF | 11.35 U/mL | [113] |
| | *Fusarium* sp. | SSF | 2.55 U/mL | [113] |
| | *Aspergillus niger*11T51A14 | SSF | 153.4 U/g | [114] |
| | *Rhizopus arrhizus* | SSF | 5.5 U/g | [115] |
| | *Candida rugosa* NCIM 3467 | SSF | 21 U/mL | [116] |
| | *Penicillumcitrinum* NCIM 765 | SSF | 16 U/mL | [116] |
| Soybean meal | *Aspergillus niger* | SSF | 25.07 U | [88] |
| Soybean bran | *Penicillium* sp. | SSF | 6.6 U/mL | [113] |
| | *Aspergillus* sp. | SSF | 11.35 U/mL | [113] |
| | *Fusarium* sp. | SSF | 2.55 U/mL | [113] |
| Corn flour | *Rhizopus arrhizus* | SSF | 1.1 U/g | [115] |
| Wheat flour | *Rhizopus arrhizus* | SSF | 4.9 U/g | [115] |
| Sunflower meal | *Rhizopus arrhizus* | SSF | 4.2 U/g | [115] |
| Oat bran | *Rhizopus arrhizus* | SSF | 3 U/g | [115] |
| Red gram husk | *Aspergillus niger* MTCC 872 | SSF | 7.17 U/g | [117] |
| Rice husk | *Aspergillus niger* MTCC 872 | SSF | 4.2 U/g | [117] |
| Sugarcane bagasse | *Rhizopus microsporus*CPQBA 312-07 DRM | SSF | 264 U/g | [118] |
| Almond meal | *Rhizopus oligosporus*ISU-16 | SSF | 81.22 U/g | [119] |

## 8. Industrial Effluents

On a global scale, the main causes of surface and groundwater pollution are the byproducts of various industries such as food processing as well as petrochemical, fertilizer, chemical and pesticide production, energy and power, mining and smelting, among others. Industrial companies generate waste in the form of solids, liquids or gases. Most of the solid waste and wastewater produced are discharged into the soil and water, and thus lead to serious environmental and economic problems [120]. A high level of water pollution causes an increase in the biological oxygen demand, chemical oxygen demand, total dissolved solids, total suspended solids, and toxic metals such as Cd, Cr, Ni, Pb, and makes water unsuitable for drinking, irrigation and aquatic life [121]. Wastewater from oil refineries, slaughter houses, and the dairy industry have proven to be a potential substrate facilitating lipase production because of the high lipid content. Lipids such as fat oils and greases are the main organic compounds in industrial and municipal wastewater. Lipids form a layer on the water surface and decrease the oxygen transfer rate [122]. Palm oil production generates waste-like oil palm trunks, oil palm fronds, empty fruit bunches, palm pressed fibers, palm kernel shells and fibrous material such as palm kernel cake and a liquid discharge of palm oil mill effluent (POME). Palm oil mill effluent contains high concentrations of such compounds as oil and grease, suspended solids, organic matter, and plant nutrients [123]. During the production of 1 ton of crude palm oil, more than 2.5 tons of POME are produced [124]. The production of olive oil mainly generates solid residues such as olive oil cakes, and liquid residues such as olive mill wastewater. Both types of waste have a high organic value, and the quantity and quality of these residues are dependent on several factors, for example, climatic conditions, cultivation methods, the region of origin, the type of olives, and the extraction technology [125]. Also, olive mill wastewater contains a high concentration of lignins and tannins, which give it a characteristic dark color. It also contains phenolic compounds and long-chain fatty acids, which are toxic to microorganisms and plants [126].

Salihu et al. [127] evaluated the feasibility of using ten different microorganisms incubated in palm oil mill effluent-based medium, and their potential to produce lipases.

Different concentrations of peptone, malt extract, urea, NH$_4$Cl and olive oil were optimized to maximize the lipase production in palm oil mill effluentsupplemented medium. The highest lipase activity was recorded by *Candida cylindracea* ATCC 14830. Medium supplementation by NH$_4$Cl and olive oil led to an enzyme activity of 2.07 U/mL. After the optimization of *Candida cylindracea* ATCC 14830, a maximum lipolytic activity of 4.02 U/mL was observed in palm oil mill effluent supplemented with peptone (0.5% *w/v*), malt extract (0.4% *w/v*) and olive oil (0.5% *v/v*). Another study, conducted by Moftah et al. [128], examined the application of solid (olive oil cakes) and liquid (olive mill wastewater) waste from the olive oil processing industry, which were evaluated as substrates for *Yarrowialipolytica* using both submerged and solid-state fermentation. The highest lipolytic activity of 850 IU/dm3 was achieved after 4 days of submerged fermentation in supplemented olive mill wastewater by the addition of ammonium sulphate (0.6% *w/v*), yeast extract (0.1% *w/v*), maltose (0.5% *w/v*), olive oil (0.3% *w/v*) and peptone I (0.1% *w/v*). Lipase production under solid-state fermentation reached a value of 40 IU/g of substrate. The objective of the study conducted by Hermansyah et al. [94] was to obtain the optimum value of lipase activity produced by the cultures of *Pseudomonas aeruginosa*, using palm oil mill effluent as a substrate through the submerged fermentation method, and to obtain a dry extract of lipase. The lipase activity level of crude extract obtained from *Pseudomonas aeruginosa* cultures, using palm oil mill effluent as the basal medium, reached a maximum value of 1.327 U/mL. The optimum values of the lipase activity unit were gained when 3% (*v/v*) of inoculum, 0.9% (*m/v*) of peptone, 0.4% (*v/v*) of olive oil, 4 mM of Ca$^{2+}$ ions, and 0.9% of Tween 80 were added into the medium and fermented for 96 h. Another study, conducted by Brozzoli et al. [52], demonstrates the application of olive mill wastewater and its potential to produce lipase through the activity of *Candida cylindracea* in a stirred tank reactor. The maximum lipase activity reached 21.6 U/mL through the activity of *Candida cylindracea* NRRL Y-17506 grown in a stirred tank reactor containing olive mill wastewater-based medium with added olive oil (3.0 g/L), under optimized pH and agitation conditions (pH < 6.5 and variable regime, respectively) for 175 h. D'Annibale et al. [129] studied the screening of 12 microorganisms using olive mill wastewater-based medium and their potential to produce lipase. All of the strains were able to grow in the undiluted olive mill wastewater, producing extracellular lipase activity. *Candida cylindracea* NRRL Y-17506 showed the highest degree of lipase activity of 9.23 IU/mL in olive mill wastewater supplemented with NH$_4$Cl (2.4 g/L) and olive oil (3.0 g/L). Kumar et al. [130] reported the discovery of a newly isolated fungal strain of *Penicillium chrysogenum* for the fermentation process. Solid-state fermentation (SSF) was carried out using grease waste and Czapek-dox medium, supplemented with wheat bran. After the optimization of the inoculum, the age, size, temperature, pH, incubation time, and grease and wheat bran ratio, the lipase activity was enhanced by up to 46 U/mL at 30 °C and pH 7.0 with grease waste, wheat bran and Czapek-dox media in the ratio of 1:1:2 (*w/w/v*). Ertugrul et al. [131] used a mixture of olive mill wastewater, whey and triolein as a medium to produce intracellular and extracellular lipase with 17 strains. During the strain possessing, the highest lipase activity was identified for *Bacillus* sp. The fermentation media was composed of 20% (*v/v*) whey + 1% (*v/v*) triolein and 20% (*v/v*) whey + 1% (*v/v*)olive mill wastewater in addition to 5.0 g/L peptone and 3.0 g/L yeast extract, and the fermentation process was carried out for 64 h at 30 °C with a pH of 6.0. The medium contained 20% whey and 1% triolein and was found to produce the highest lipase activity. The cultivation of *Bacillus* sp. resulted in extracellular and intracellular lipase activities of 15 U/mL and 168 U/mL, respectively. The conditions of the production of lipase using industrial effluents are presented in Table 6.

**Table 6.** Production of lipase using industrial effluents.

| Substrates | Microorganisms | Type of Fermentation | Maximum Lipase Activity | Source |
|---|---|---|---|---|
| Palm oil mill effluent | *Penicillium* spp. E-01PC; *Penicillium* sppE-04PC; *Penicillium* spp. E-06PC; *Penicillium* spp. E-07PC; *Penicillium* spp. E-08PC; *Penicillium* sppE-10PC *Mucor hemalis*; *Rhizopusoryzae*; *Candida cylindracea* ATCC 14830; *Penicillium citrinum* ATCC 42799 | SmF | 4.02 U/mL (*Candida cylindracea* ATCC 14830) | [127] |
| Olive mill wastewater; Olive oil cake | *Yarrowialipolytica* NRRL Y-1095 | SmF | 850 IU/dm$^3$ | [128] |
| Palm oil mill effluent | *Pseudomonas aeruginosa* B2290 | SmF | 1.327 U/mL | [94] |
| Olive mill wastewater | *Candida cylindracea* NRRL Y-17506 | SmF | 21.6 U/mL | [52] |
| Olive mill wastewater | *Geotrichumcandidum* NRRLY-552; *Geotrichumcandidum* Y-553; *Rhizopus arrhizus* NRRL 2286; *Rhizopus arrhizus* ISRIM 383; *Rhizopus oryzae* NRRL 6431; *Aspergillus oryzae* NRRL 1988; *Aspergillus oryzae* NRRL 495; *Aspergillus niger* NRRL 334; *Candidacylindracea* NRRL Y-17506; *Penicillium citrinum* NRRL 1841; *Penicillium citrinum* NRRL 3754; *Penicillium citrinum* ISRIM 118 | SmF | 9.23 IU/mL (*Candida cylindracea* NRRL Y-17506) | [129] |
| Grease waste; Wheat bran | *Penicilliumchrysogenum* | SSF | 46 U/mL | [130] |
| Olive mill wastewater; Whey | *Bacillus* sp. | SmF | 15 U/mL (extracellular) 168 U/mL (intracellular) | [131] |

## 9. Conclusions

Lipases are ubiquitous enzymes with an important physiological role and are capable of catalyzing various reactions, which makes them very useful biocatalysts for industrial applications. Agro-industrial wastes and residues are rich in nutrient composition and bioactive compounds. Microorganisms such as bacteria, filamentous fungi and yeast can be deployed for the production of lipases using agricultural wastes. Lipases produced by microorganisms, with the use of renewable agro-industrial residues and wastes, may be a viable alternative to commercially available lipases. Agro-industrial residues and waste, such as peels, oil-cakes, bagasses, husks, seeds, and leaves, can be used as a solid support in solid-state or submerged fermentation processes for the production of lipases and a range of significantly beneficial enzymes or compounds.The application of optimal process conditions for productive microorganisms (*Aspergillus niger*, *Rhizopus microspores*, *Streptomyces indiaensis*),in combination with properly selected oil-rich agro-industrial waste, giveslipase activity even above 200 U/g. Thus, using renewable sources and highly effective microorganisms to produce important and useful compounds can help to reduce their production costs and recycle waste, thereby making it more eco-friendly.

**Author Contributions:** Conceptualization, T.S. and J.C.; writing—original draft preparation, T.S.; writing—review and editing, J.C., M.P., M.F.; visualization, T.S.; supervision, J.C., M.P. All authors have read and agreed to the published version of the manuscript.

**Funding:** This research was funded by the Ministry of Science and Higher Education, contract number DWD/3/51/2019 of 22 October 2019.

**Acknowledgments:** The work was financed by the Ministry of Science and Higher Education under the contract no. DWD/3/51/2019 of 22 October 2019.

**Conflicts of Interest:** The authors declare no conflict of interest. The funders had no role in the design of the study; in the collection, analyses, or interpretation of data; in the writing of the manuscript, or in the decision to publish the results.

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
