# Peer review of "Various Perspectives on Microbial Lipase Production Using Agri-Food Waste and Renewable Products"

_agriculture, doi:10.3390/agriculture11060540_

Round 1

Reviewer 1 Report

Authors reviewed microbial lipase production using agri-food waste and renewable products. The lipase characteristics and its application were description. However, the microbial lipase production using agri-food waste was not well organized and weak in the manuscript. The literatures about microbial lipase production were only explained in a narrative way, without classification and planning, especially in the latter part. The comment of this manuscript is reconsider after major revision. The following specific comments should be considered while revising the manuscript.

Major Promlem

  1. Table 4 does not have a good classification, just put a lot of references in the table. For example, there can be only one type of substrates in the first column, and there can be many kinds of microorganisms in the second column. Please refresh the table.
  2. In addition, the unit of activity is different for each reference, how did they compare with each other?
  3. The problems in Table 5 are the same as those in Table 4

Minor

  1. L.38, Add a comma before “respectively”.
  2. L.89, 1,3 specific lipases.. S should be capital.
  3. L. 90, It should be 2,3 or 2,3 diacylglycerols.
  4. L. 92, This is a special case, so write clearly. Please rewrite as below:

Fatty acid specific lipases hydrolyse esters, which have long-chain fatty acids with double bonds on C-9, such as the lipase from Geotrichum candidum [20].

  1. L. 116-127, Some examples should be added to the lipase-catalyzed reaction.

For example, L.118

Esterification is a double displacement reaction between alcohols and carboxylic acids resulting in esters and water [cite ref.].

Esterification

Kuo, C.H., Chen, H.H., Chen, J.H., Liu, Y.C., Shieh, C.J. 2012. High yield of wax ester synthesized from cetyl alcohol and octanoic acid by Lipozyme RMIM and Novozym 435. International Journal of Molecular Sciences. 13, 11694-11704.

Alcoholysis

Huang, S.M., Huang, H.Y., Chen, Y.M., Kuo, C.H., Shieh, C.J. 2020. Continuous production of 2-phenylethyl acetate in a solvent-free system using a packed-bed reactor with Novozym® 435. Catalysts. 10(6), 714.

Acidolysis

Kuo, C.H., Huang, C.Y., Lee, C.L., Kuo, W.C., Hsieh, S.L., Shieh, C.J. 2020. Synthesis of DHA/EPA ethyl esters via lipase-catalyzed acidolysis using Novozym® 435: A kinetic study. Catalysts. 10, 565.

Interesterification

Lee, J. H., Son, J. M., Akoh, C. C., Kim, M. R., Lee, K. T. 2010. Optimized synthesis of 1, 3-dioleoyl-2-palmitoylglycerol-rich triacylglycerol via interesterification catalyzed by a lipase from Thermomyces lanuginosus. New biotechnology, 27(1), 38-45.

  1. L. 127-130, This paragraph is unclear.
  2. L.144-146, the sentence is not clear, please rewrite the sentence.

The use of an encapsulated enzyme contains a lipase (Palatase M from Novozymes) and a protease can be used to obtain a full-flavoured cheese without a bitter taste [33].

  1. L.191, The words “production” in the file are illegible, please check.
  2. L.273, There are two spaces in this sentence.
  3. Use the red arrow on the Figure 4,a,b,c to point out the mentioned in the text.
  4. L.306, Gc is also a commonly used method, it should be mentioned. Here has a related reference.

Kulkarni, N., & Gadre, R. V. (1998). Simple gas chromatography method for lipase assay. Biotechnology techniques, 12(8), 627-628.

  1. Table3. This table contains only the composition of agro-industrial residues and wastes. The main purpose of this review is to use agricultural waste to produce lipase. Please put the references for lipase production from microorganisms used these agro-industrial residues and wastes.
  2. The genus and species names of the bacteria should be in italics. Please check whole text.

Reviewer 2 Report

The authors present an interesting review regarding the various perspectives on microbial lipase production using agri-2 food waste and renewable products. The authors should consider the below comments in order to improve their paper.

  1. Line 20 please correct at the keywords "solid-stae" with solid-state.
  2. In section 1, it would be useful to provide an image with the structure of the enzymes.
  3. Please clarify line 127: ...[8]. hors should...:
  4. Equation should be numbered and cited in the main text.
  5. Considering that it is a review article the authors should compare the fermentation techniques based on numerical values which can be provided in table 2. Also please point out the values of the key operating parameters of the fermentation processes. Such a presentation and discussions of the literature data would give a much comprehensive overview of the fermentation techniques.
  6. Please complete the title of section 5: for lipases production.
  7. Conclusion section should be detailed and backed-up with numerical values.

Round 2

Reviewer 1 Report

The article has been revised according to the suggestions, and Table 3, 4, and 5 are well organized now, match with the topic and focus on agri-food waste and renewable products. This manuscript can be accepted after the suggestion below are revised.

  1. Use commas to separate sentences in L. 12-15.

2.L. 93. It should be 2,3 or 1,2 diacylglycerols…….

Author Response

We would like to thank the Reviewer for critical review of our manuscript. We have corrected the paper according to Reviewer's suggestions. All corrections are introduced in green color.